

# *SPANXN2* functions a cell migration inhibitor in testicular germ cell tumor cells

Fang Zhu[1], Hao Bo[1,2], Guangmin Liu[1], Ruixue Li[1], Zhizhong Liu[1,3] and Liqing Fan[1,2]

[1] Institute of Reproductive & Stem Cell Engineering, School of Basic MedicalScience, Central South University, Changsha, Hunan, China
[2] Clinical Research Center for Reproduction and Genetics in Hunan Province, Reproductive and Genetic Hospital of CITIC-Xiangya, Changsha, Hunan, China
[3] Hunan Cancer Hospital, Department of Urology, The Affiliated Cancer Hospital of Xiangya School of Medicine of Central South University, Changsha, Hunan, China

Corresponding authors
Zhizhong Liu,
liuzhizhong@hnca.org.cn
Liqing Fan, liqingfan@csu.edu.cn

## ABSTRACT

**Background**. *SPANX* family members are thought to play an important role in cancer progression. The *SPANXN2* is a gene expressed mainly in normal testis, but its role in testicular germ cell tumors (TGCTs) has yet to be investigated. TGCT is one of the most common solid tumors in young men and is associated with poor prognosis; however, effective prognostic indicators remain elusive. Therefore, we investigated the role of *SPANXN2* in TGCT development.

**Methods**. *SPANXN2* expression levels were validated by quantitative real-time polymerase chain reaction (qRT-PCR) analyses of 14 TGCT samples and five adjacent normal tissue samples. *SPANXN2* was transiently overexpressed in TGCT cells to study the consequences for cell function. The effects of *SPANXN2* on cell migration were evaluated in transwell and wound healing assays. The effects on cloning ability were evaluated in colony formation assays. MTT assays and cell cycle analysis were used to detect the effects of *SPANXN2* on cell proliferation. The expression levels of EMT- and AKT-related proteins in cells overexpressing *SPANXN2* were analyzed by Western blotting.

**Results**. Compared with adjacent normal tissues, the Gene Expression Profiling Interactive Analysis database showed *SPANXN2* expression was downregulated in TGCTs which was consistent with the qRT-PCR analysis. *SPANXN2* overexpression reduced cell migration and colony formation capability and downregulated expression of EMT- and AKT-related proteins, Vimentin, Snail, AKT, and p-AKT.

**Conclusion**. Our results suggest that *SPANXN2* regulates TGCT cell migration via EMT- and AKT-related proteins although its role in the occurrence and development of TGCT remains to be fully elucidated.

## INTRODUCTION

Testicular germ cell tumor (TGCT) is one of the most common solid tumors in young men aged between 15 and 40 years (*Rijlaarsdam & Looijenga, 2014*). The incidence of TGCTs has increased globally in recent years, to approximately 1.3/100,000 men in China and 7.8/100,000 men in Western Europe (*Shanmugalingam et al., 2013*; *Le Cornet et al., 2014*; *Znaor et al., 2014*). Histologically, there are two main subtypes of TGCTs, i.e., seminoma and non-seminoma (*Lobo et al., 2018*). Both seminoma and non-seminoma carry risks of vascular invasion and the development of distant metastases (*Yilmaz et al., 2013*). Seminoma is distinctive in its sensitivity to cisplatin-based chemotherapy and radiation, and 20% of them are destined to relapse (*Schmidberger et al., 1997*; *Classen et al., 2003*; *Facchini et al., 2019*). Approximately 30% of patients with clinical first-stage non-seminoma have subclinical metastases in retroperitoneal lymphatic stations and lung metastasis after chemotherapy (*Yilmaz et al., 2013*; *Albers et al., 2003*). Moreover, the occurrence and treatment of TGCTs are associated with impairment of the sexual function, fertility, and quality of life, especially in young patients (*Van den Belt-Dusebout et al., 2007*; *Albers et al., 2015*). Tumor progression and poor prognosis are usually caused by metastasis; therefore, clarification of the molecular mechanisms underlying the pathogenesis and development of TGCTs is needed.

The *SPANX* multigene family is a representative cancer-testis antigen, which has two subfamilies: *SPANX-A/D and SPANX-N* (*Whitehurst, 2014*; *Kouprina et al., 2004*; *Kouprina et al., 2007a*; *Kouprina et al., 2007b*). The *SPANX-N* subfamily consists of five members, *SPANXN1* (-*N2*, -*N3*, -*N4*, and -*N5*). Several studies of *SPANX* family members in breast cancer, colorectal cancer, and lung adenocarcinoma showed that their relationship with metastasis and poor prognosis in cancers (*Chen et al., 2010*; *Maine et al., 2016*; *Hsiao et al., 2016*). However, the role of *SPANX* family members in TGCTs has not yet been described (*Kouprina et al., 2007a*; *Kouprina et al., 2007b*). The *SPANXN2* gene is localized on chromosome Xq27, a region of susceptibility gene localization for TGCT and prostate malignancy (*Rapley et al., 2000*; *Kouprina et al., 2007a*; *Kouprina et al., 2007b*; *Lutke et al., 2006*). In this study, we explore the role of *SPANXN2* in TGCT progression to understand the importance of the *SPANXN2* gene in TCGT and provide insights into the role of*SPANXN2* in the progression of TGCT.

In our study, the effect of *SPANXN2* on TGCTs progression investigated *in vitro*. Our result showed that *SPANXN2* inhibited TGCT cell migration, indicating that *SPANXN2* is an inhibitor of tumor metastasis.

## MATERIALS & METHODS

### Human testicular samples

The adjacent normal testicular tissue and TGCTs tissue samples used in this study were obtained from the Affiliated Cancer Hospital of Central South University (Changsha, China). Five adjacent normal tissue samples had been removed during para-testicular tumor surgery and the TGCT tissue samples were obtained from 11 testicular seminomas and three non-seminomas. Fresh tissues were collected and frozen in liquid nitrogen for

storage at −180 °C. All the tissues were confirmed by histopathological examination. The patients provided written informed consent to tissue sample collection, which was performed with the authorization of the Ethics Committee of Central South University (Approve No.: LLSB-2017-002).

## Quantitative RT-PCR

The total RNA was extracted using TRIzol Reagent (Invitrogen, USA). The amount and purity of each RNA sample were quantified by Agilent2100 (Agilent, Wilmington, DE, USA). The cDNA was synthesized from 1 μg RNA using the Transcriptor First Strand cDNA Synthesis Kit (Roche, USA). The real-time PCR system (LightCycler480, Roche, USA) was used to measure the relative expression level of the *SPANXN2* gene and the β-*actin* was used as the housekeeping gene for normalization. Amplification was performed with the following thermo-cycling conditions: initial denaturation at 95 °C for 5 min, followed by 45 cycles of 95 °C for 10s and 60 °C for 10 s, and a final extension at 72 °C for 10 s. The LightCycler480 software was used to analyze the threshold cycle (CT) values and the $2^{-\Delta\Delta CT}$ method was used to evaluated relative gene expression. The gene-specific primers used were as follows:

*SPANXN2* forward: 5′-GTGTATTACTACAGGAAGCATACG-3′;
reverse: 5′-CTCCTCCTCTTGGACTGGATT-3′
β − *actin* forward: 5′-TCACCAACTGGGACGACATG-3′;
reverse: 5′-GTCACCGGAGTCCATCACGAT-3′

## Cell culture

The human TGCT cell line NCCIT was bought from the American Type Culture Collection (ATCC, VA, USA), and the human TGCT cell line TCAM-2 was obtained from Dr. Yuxin Tang (*Peng et al., 2019*; *Gan et al., 2016*). NCCIT cells were cultured in RPMI-1640 medium (GIBCO, USA), and TCAM-2 cells were cultured in Dulbecco's Modified Eagle's Medium (DMEM, GIBCO, USA). All cells were cultured in medium containing 10% fetal bovine serum (FBS, GIBCO, USA), 100 U/ml penicillin and 100 μg/ml streptomycin (GIBCO) and were incubated at 37 °C under 5% $CO_2$.

## Cell transfection

The sequence of *SPANXN2* was cloned into the CMV-MCS-DsRed2-SV40-Neomycin-GV147 vector. Cells were cultured as described above and divided into negative control (NC) and test (SPANXN2) groups and transfected with the GV147 empty vector (NC) and the GV147 vectors expressing *SPANXN2*, respectively. Briefly, cells were seeded in a 6-well culture plate ($5\times10^5$ cells/well) and transfected at 70% confluence. Cells were transfected with 2.5 μg of the GV147 empty vector and 2.5 μg of the GV147 vectors expressing *SPANXN2* using DNA Lipofectamine 3000 (Invitrogen, USA) according to the manufacturer's instructions. Cells were harvested 36 h or 48 h after transfection for use in the following experiments.

## Plate colony formation assay

Cells were seeded in a 6-well plate (300 cells/well) and incubated at 37 °C under 5% $CO_2$ for 12–14 days when most single-cell colonies consist of >50 cells. The cells were washed three

times with phosphate-buffered saline (PBS) and fixed in 4% paraformaldehyde for 30 min and then stained for 15 min with 0.5% crystal violet. After staining, colonies containing >50 cells were observed under a microscope and the colony number (a single colony with >30 cells) was counted using Adobe Photoshop CC 2018.

## MTT assay

The proliferation of TGCT cells was evaluated by MTT assay. The cells were seeded into 96-well plates at $5 \times 10^3$ cells/ml (200 µl/well). After 6 h, 20 µl MTT solution (5 mg/ml) (3-(4, 5-dimethylthiazol-2-yl)-2, 5-diphenyltetrazolium bromide, MTT) reagent (Sigma Chemicals, St Louis, MO, USA) was added to the 96-well plates. After incubation for 6 h at 37 °C, the supernatant was discarded from each well and 200 µl DMSO was added to each well. The absorbance at 492 nm was measured using a Nanodrop 1000 spectrophotometer (Thermo Fisher, USA). This process was repeated every 24 h for 1, 2, 3, and 4 days.

## Flow cytometric cell cycle analysis

At 48 h after transfection, cells were first enzymatically dissociated into single cells by 0.05% trypsin and then fixed with 75% cold ethanol in PBS overnight at 4 °C. Second, the cells were washed with PBS three times. Finally, the cells were double-stained with a propidium iodide solution (PI, 50 µg/mL) and treated with RNase (1 mg/mL) at 37 °C for 30 min in the dark. Subsequently, samples were analyzed using an Accuri C6 flow cytometer (BD) with the Accuri software. For each sample, data from approximately 20,000 cells were acquired in list mode using logarithmic scales. FlowJo7.6.2 was used to analyze the cell cycle distribution according to the guideline.

## Wound healing assay

Cells were seed in a 6-well culture plate ($5 \times 10^5$ cells/well) and the next day, transfected at 70% confluence. At 48 h after transfection, we drew three straight lines with an average interval in the layer of cells with a 100 µl of the tip. Cells were then washed three times with PBS to remove the scraped suspended cells. Cells were then washed cultured in medium containing 2% fetal bovine serum, 100 U/ml penicillin and 100 µg/ml streptomycin at 37 °C under 5% $CO_2$. The wound healing areas were recorded by observation under a microscope at 0 h, 24 h and 48 h, and Image-Pro Plus 6.0 software was used to analyze the width of wound healing.

## Transwell migration assay

Transwell Cell Culture Inserts (8 µm pore size, FALCON, USA) were used to measure cell migration capacity in 24-well plates. In brief, TCAM-2 and NCCIT cells ($2 \times 10^4$ cells in 200 µl of 2% FBS medium) were added to the upper chamber, and 800 µl 15% FBS medium was added to the lower chamber. After incubation at 37 °C under 5% $CO_2$ for 48 h, the cells on the upper surface were wiped with cotton swabs, while the cells that migrated through the filter pores were fixed in 4% paraformaldehyde for 30 min. After staining with 0.5% (w/v) crystal violet (Sigma) in PBS (GIBCO) for 15 min, cells were observed and photographed under an inverted microscope. Cells were counted in five randomly selected fields of view.

## Western Blot analysis

Cells were collected and lysed with RIPA lysis buffer (Co Win Biosciences, China) according to the manufacturer's instructions, and proteins were harvested. Protein quantification was performed using a BCA Protein Quantification Kit (Thermo, USA), and samples were adjusted to the same concentration. After boiling in 6× Loading Buffer for 5 min, the proteins were separated by SDS-PAGE and transferred to polyvinylidene fluoride (PVDF) membranes (Millipore, USA). The membrane was incubated overnight at 4 °C with the appropriate primary detection antibodies: GAPDH (Sigma, 1:1,000), α-Tubulin (Sigma, 1:600), Snail (Cell Signaling Technology, 1:1,000), and Vimentin (Cell Signaling Technology, 1:1,000); AKT (ProteinTech, 1:1,000) and p-AKT (ProteinTech, 1:1,000). After washing three times with TBST for 5 min, the membrane was incubated with secondary antibodies (Cell Signaling Technology, 1:1,000) for 1 h at 37 °C. After washing three times with TBST for 15 min each time, the protein bands were visualized by fluorography using an enhanced chemiluminescence system (Millipore, USA); the expression levels of α-Tubulin and GAPDH were used as internal references.

## Statistical analysis

All assays were repeated three times and all samples were analyzed in triplicate. The data were expressed as the mean ± SD or mean ± SEM of at least three independent experiments. Student's $t$-test or a two-way ANOVA was used to determine the significance of differences between groups, and $P < 0.05$ was deemed statistically significant. Curve fitting analyses were performed with GraphPad Prism Software5.0 (GraphPad Software, USA).

# RESULTS

## Validation of *SPANXN2* expression in TGCTs

To verify the reliability of differential *SPANXN2* expression, we analyzed its expression in TGCTs in the GEPIA database (*Tang et al., 2017*) (http://gepia.cancer-pku.cn/index.html), and the results showed that *SPANXN2* was significantly downregulated in TGCT tissues (Fig. 1A). Subsequent qRT-PCR analysis of TGCT samples and normal samples also confirmed the differential expression of *SPANXN2* (Fig. 1B). These results revealed that TGCT tissues express a lower level of *SPANXN2* relative to those detected in normal tissues.

## Transfection of TGCT cell lines

As shown in Figs. 2A, 2B, most cells (>75%) showed red fluorescence that confirmed transfection with plasmids targeting *SPANXN2*, or an empty-plasmid control. Quantitative RT-PCR was used to detect the relative expression of *SPANXN2* in these two transfected TGCT cell lines (TCAM-2, NCCIT), using β − *actin* as a reference (Figs. 2C and 2D). The results confirmed the relative overexpression of *SPANXN2* in the*SPANXN2* group compared with that in the NC group, laying the foundation for the study of its function.

## *SPANXN2* upregulation inhibited colony formation ability with no effect on cell proliferation

The effect of *SPANXN2* on the ability of TGCT cells to form colonies was investigated in colony formation assays. The colony formation ability of TGCT cells overexpressing

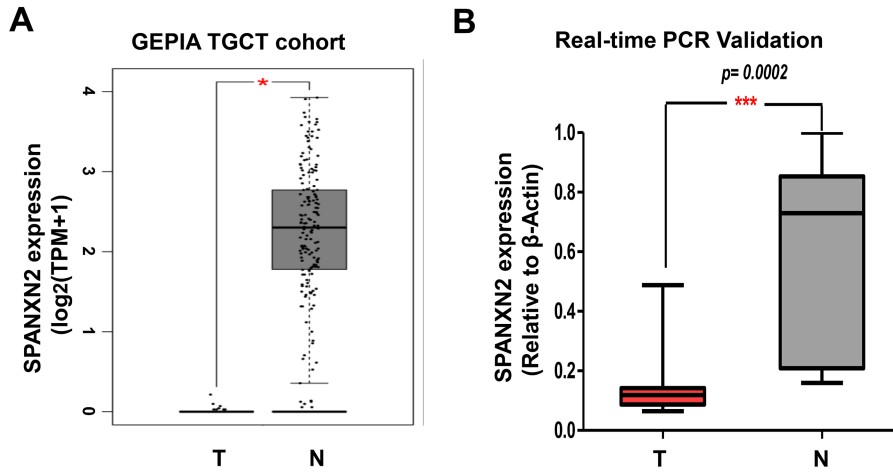

**Figure 1** *SPANXN2* is downregulated in TGCT relative to normal tissues. (A) The GEPIA database shows the mRNA levels of *SPANXN2* expression was significantly downregulated in TGCT tissues compared to normal tissues. TPM=Transcription per million. (B) Independent quantitative RT-PCR validation of TGCT tissue and normal samples. "T" indicates TGCT specimens, "N" indicates normal specimens.

*SPANXN2* was significantly decreased compared with that of the cells in the NC group ($P < 0.05$, Figs. 3A and 3B).

To investigate the effect of *SPANXN2* overexpression on TGCT cell proliferation. NCCIT and TCAM-2 cells were transfected with the *SPANXN2* plasmid and their negative control plasmid. The MTT assay results revealed no significant difference in the cell proliferation rates of the NC and *SPANXN2* groups ($P > 0.05$, Figs. 3C and 3D). Furthermore, flow cytometric analysis carried out 48 h after transfection showed that there were no significant differences in the cell cycle distribution of TGCT cells overexpressing *SPANXN2* compared with the cells in the NC group ($P > 0.05$, Figs. 4A–4F). In addition, no marked cell apoptosis was observed at 24 h and 48 h after transfection with the *SPANXN2* plasmid. These results indicated that *SPANXN2* overexpression inhibited the colony formation ability of TGCT cells, but has no significant effect on the cell cycle distribution and apoptosis.

### *SPANXN2* negatively regulates TGCT cell migration

The effect of *SPANXN2* overexpression on the migration of TGCTs cells was investigated in wound healing and Transwell assays. As shown in Fig. 5, the wound healing gap in the NC group was significantly smaller than that in the *SPANXN2* group. At 48 h after transfection, there were significantly fewer migrated cells in the *SPANXN2* group compared with that in the NC group ($P < 0.05$, Figs. 6A and 6B). The results indicated that *SPANXN2* overexpression inhibited TGCT cell migration.

### *SPANXN2* influences the expression of EMT- and AKT-related proteins

EMT describes the transition of epithelial cells into mesenchymal cells, some molecules of which will be changed during the process of metastasis of many tumors. To explore how *SPANXN2* inhibited TGCT cell migration, the expression levels of EMT-related proteins

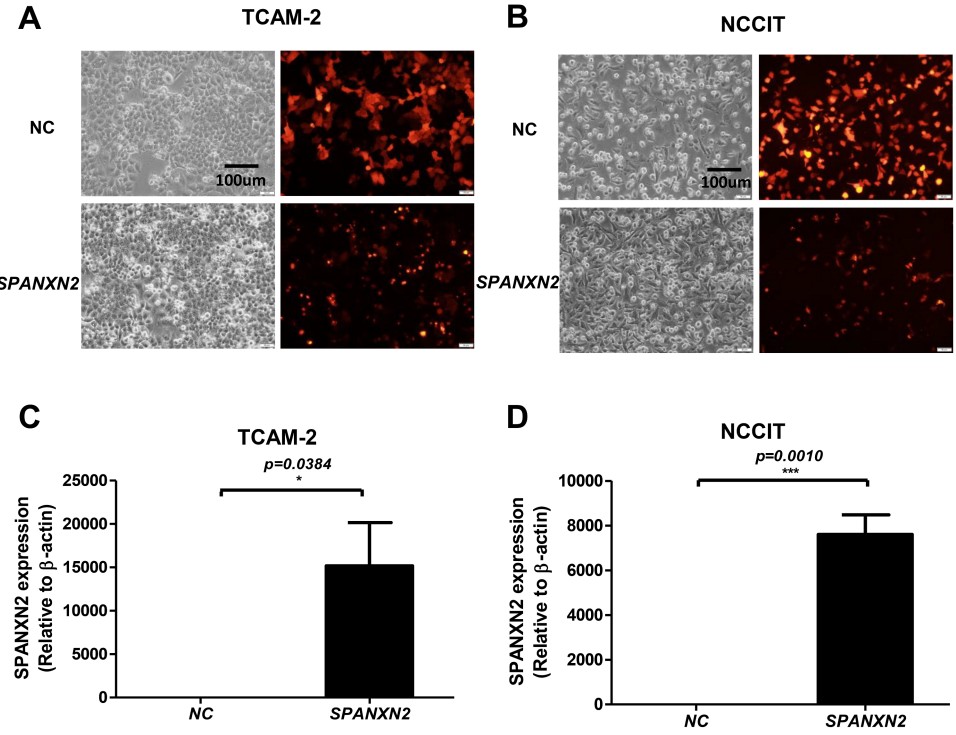

**Figure 2** **Transfection of TGCT cell lines.** (A, B) Plasmid transfection efficiency evaluated by fluorescence microscopy. (C, D) Relative expression of *SPANXN2* in TCAM-2 and NCCIT cells was measured by qRT-PCR in the NC and SPANXN2 groups using $\beta$-actin as a reference. Data represent the mean $\pm$ SEM and * $P < 0.05$, ** $P < 0.01$, *** $P < 0.001$. Scale bar in A and B, 100 $\mu$m.

were detected by western blot analysis. These results revealed that the expression levels of the EMT related proteins, Vimentin, and Snail were decreased (Fig. 6C), suggesting that *SPANXN2* regulates the EMT-related proteins. Similarly, the AKT-related molecules were also detected. AKT and p-AKT protein expression levels were lower after *SPANXN2* overexpression (Fig. 6D). These results showed that *SPANXN2* overexpression regulated EMT- and AKT-related proteins in TGCT cells.

## DISCUSSION

TGCTs are one of the major male reproductive tumors in young men (*Rijlaarsdam & Looijenga, 2014*). The clinical treatment of TGCTs is mainly orchiectomy supplemented with chemotherapy and radiotherapy (*Greene et al., 2010*). Data show that the incidence of TGCTs has increased globally in recent years (*Shanmugalingam et al., 2013*; *Le Cornet et al., 2014*; *Znaor et al., 2015*). Despite encouraging progress in the diagnosis and treatment, TGCT patients still have a high risk of relapse with poor prognosis (*Qin et al., 2020*) . The mechanisms underlying the development of TGCT have yet to be fully elucidated (*Kalavska et al., 2017*).

The results of our analysis of clinical TGCT specimens were consistent with those of the GEPIA database, in which *SPANXN2* expression was lower in TGCTs at the mRNA

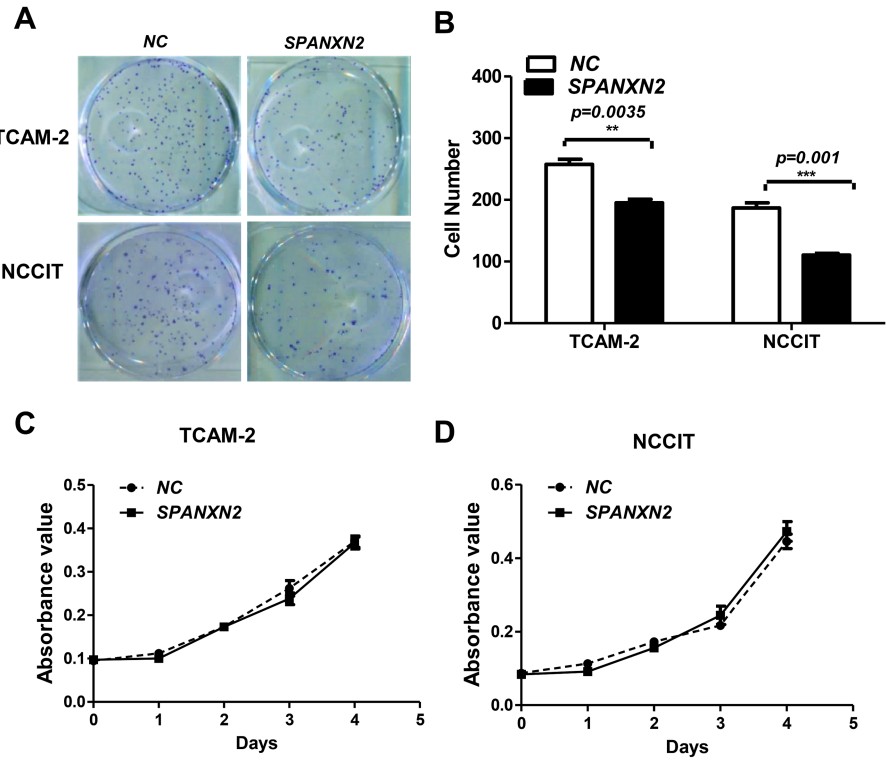

**Figure 3** *SPANXN2* upregulation demised the colony formation ability but had no effect on cell pro-liferation. (A) Colony formation ability was evaluated in colony formation assays. (B) Relative cell colony formation rate; * $P < 0.05$, ** $P < 0.01$ compared with the NC group. (C, D) Cell proliferation was evaluated in MTT assays.

level. Studies of the role of non-coding RNAs in TGCTs have been conducted. A profiling study of the small RNA populations in TGCTs showed the global loss of piwi-interacting RNA *(Rounge et al., 2015)*. In a gene expression profiling study of TGCT samples, *Siska et al., (2017)* demonstrated that activated T-cell infiltration and PD-1/PD-L1 interaction are closely correlated with seminoma histology and good prognosis. To our knowledge, *in vitro* studies of the role of *SPANXN2* in TGCT have not yet been reported and it is not clear if *SPANXN2* acts as a tumor suppressor gene or oncogene in TGCT. Previous studies have shown abnormal *SPANX* gene expression in many cancers, which may promote or inhibit tumor invasion *(Yilmaz-Ozcan et al., 2014; Maine et al., 2016)*. However, the role of *SPANXN2* in TGCTs progression has not yet been elucidated. The differential expression of *SPANXN2* between TGCTs tissues and adjacent normal testicular tissues suggests that *SPANXN2* plays a role in the progression of TGCT.

In our study, no significant effects of *SPANXN2* on proliferation activity and cell cycle distribution in TGCT cell lines were observed. However, we found that *SPANXN2* significantly inhibited the colony formation and migration abilities of TGCT cells. These findings suggested that the inhibitory effects of *SPANXN2* on the colony formation and

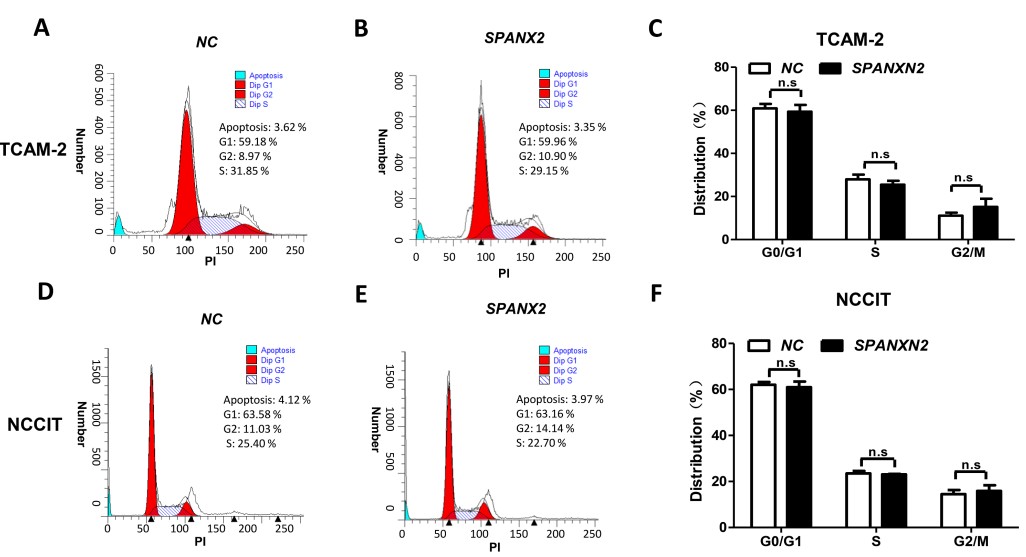

**Figure 4** **The cell cycle distribution.** (A, B, D, E) The cell cycle distribution was evaluated by flow cyto-metric analysis and FlowJo7.6.2. (C, F) Bar graphs showing the distribution of calls in each phase. n.s, no significance.

migration of TGCT cell lines are not influenced by cell proliferation and may be regulated by other pathways.

Metastasis is an important aspect of cancer progression and may lead to a poor prognosis in tumor patients. In addition to *SPANXN2*, other SPANX gene family members also play important roles in tumors metastasis. Research showed that *SPANX* expression was significantly correlated with the migration of colorectal cancer to the liver with poor prognosis (*Chen et al., 2010*). SPANXA/D was required for spontaneous metastasis of breast cancer cells to the lung and elevated SPANXA/D expression in breast cancer patient tumors correlated with poor outcome (*Maine et al., 2016*). Similar to our study, *SPANXN2* inhibited the migration of TCGT cells, suggesting that *SPANXN2* might be a tumor suppressor gene in TGCT and related to the poor prognosis of TGCT patients. Functional investigations of *SPANXN2* are restricted by the high homology among *SPANX* family members. The mechanism of *SPANXN2* dysregulation in TGCT requires further research, although our results showed that *SPANXN2* inhibited the colony formation and migration abilities of TGCT cell lines. To our knowledge, our study is the first to characterize the functional role of *SPANXN2* in TGCTs.

EMT is critical for local invasion and cell dissemination, which in cancer, are associated with tumor initiation and progression, stemness, survival, and resistance to therapy (*Mitschke, Burk & Reinheckel, 2019*). Epithelial cells convert to interstitial phenotypes, characterized by the upregulation of interstitial proteins, such as Vimentin (*Mitschke, Burk & Reinheckel, 2019*; *Jolly et al., 2019*) and EMT transcriptional drivers such as Snail (*Thiery et al., 2009*). In our study, we found that *SPANXN2* suppressed TGCT cell migration and altered the expression of the EMT-related proteins, Vimentin and Snail. These findings are in accordance with those reported by Hsiao et al., showing that *SPANXA* inhibits the

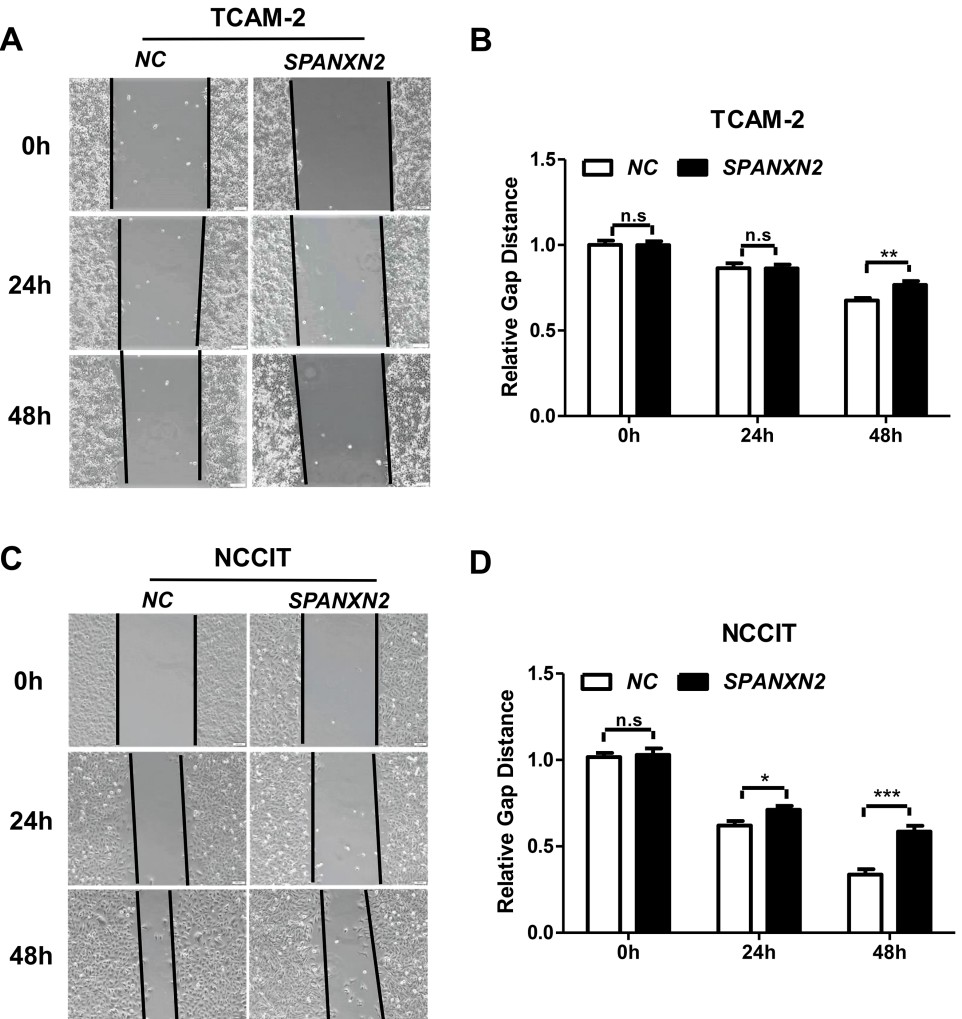

**Figure 5** **Effects of *SPANXN2* on cell wound healing.** (A, C) Image showing wound healing gap of TGCTs cells overexpressing *SPANXN2*. (B, D) The relative wound healing distance in the NC and SPANXN2 groups; * $P < 0:05$, ** $P < 0:01$, *** $P < 0:001$ compared with the NC group; n.s, no significance.

invasion and metastasis of lung cancer cells *in vitro* and *in vivo* by inhibiting the EMT pathway in lung adenocarcinoma (*Hsiao et al., 2016*). Previous studies have also indicated that CTAs, such as the SPANX protein family, activate the EMT signaling pathway in cancer stem cell-like cells, and leading to the occurrence of cancer (*Yang et al., 2015*). Therefore, we speculated that *SPANXN2* suppressed the metastasis of TGCT cells by regulating the EMT-related proteins in TGCTs.

The AKT signaling pathway is activated in many cancers, such as thyroid carcinomas (*Chen et al., 2018*). A previous study showed that activation of AKT expression drives the EMT phenotype and enhances the proliferation and invasion of seminoma (*Chen et al., 2018*). The AKT pathway is involved in cellular processes, particularly in cell proliferation and migration, and has a pivotal role in human tumorigenesis (*Chen et al., 2018*; *Minna*

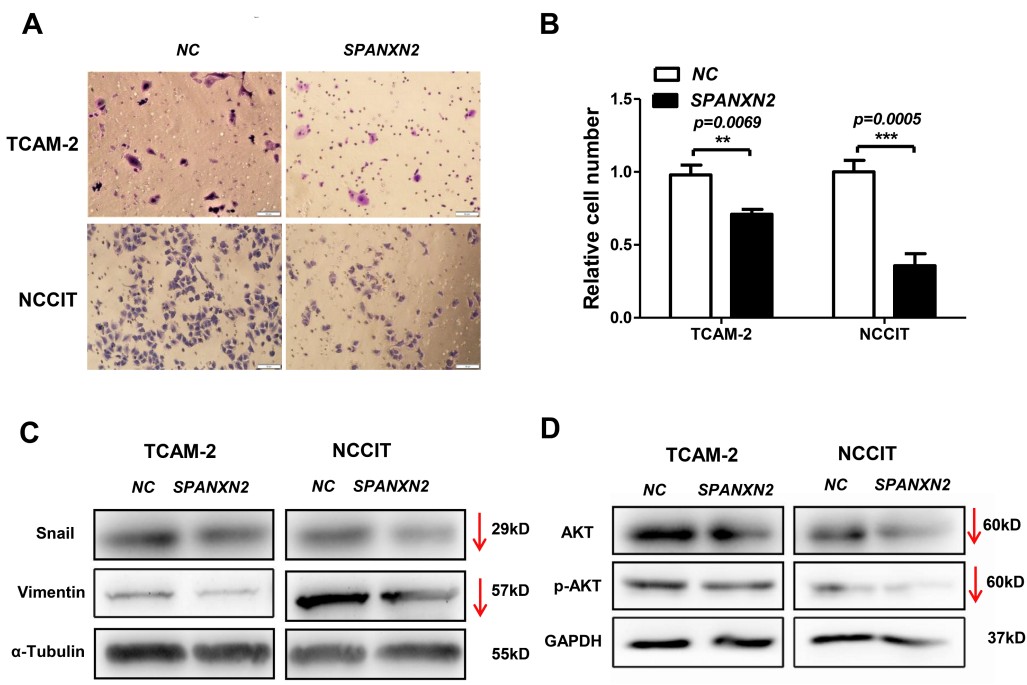

**Figure 6** **Effects of *SPANXN2* on cell migration and the related proteins of EMT and AKT.** (A) Image showing migration of TGCT cells overexpressing *SPANXN2*. (B) The relative number of migrated cells in the NC and SPANXN2 groups; \**P*<.05 compared with the NC group. (C) Western blot analysis of Vimentin and Snail proteins. (D) Western blot analysis of AKT and p-AKT proteins. Scale bar in A, 100 μm.

*et al., 2016*). In our study, the AKT proteins were downregulated in the TGCT cells overexpressing *SPANXN2*. Above all, these results suggested that *SPANXN2* suppresses tumor cell migration and colony formation via regulation of EMT- and AKT-related proteins.

Certain limitations of qRT-PCR analysis should be noted. Since clinical specimens are hard to obtain, we included biological tissues from only 14 cases, comprising a population including both metastatic and chemotherapy-refractory tumors, and all were obtained from a single institution. In addition, further studies are required to confirm the molecular mechanisms by which *SPANXN2* inhibited tumor cell migration by regulating the EMT and AKT/p-AKT signaling pathways. On the other hand, we were unable to determine the *SPANXN2* expression at the protein level is failed detected due to its sufficient homology among SPANX gene family members resulting in a lack of specificity of the SPANXN2-binding antibody. Third, the population of TGCT cells overexpressing *SPANXN2* used for clonal analysis contained some wild-type cells, which may weaken the effect of *SPANXN2* on clone formation; however, our study showed that *SPANXN2* inhibited TGCT cell clone formation, thus verifying that the observation that *SPANXN2* inhibited TGCT cell clone formation. Finally, the functions of these molecules in the progression of TGCT remain to be elucidated. According to large-scale sequencing studies of testicular germ cell tumors, there is no major high-penetrance TGCT predisposition gene (*Crockford et al.,*

*2006*; *Litchfield et al., 2018*). It is likely to be due to the involvement of many genes in the occurrence and development of TGCT.

## CONCLUSIONS

In summary, this is the first report describing the role of *SPANXN2* in TGCTs. *In vitro* studies demonstrated that *SPANXN2* significantly inhibited the migration and colony formation of TGCT cells and regulated the expression of EMT- and AKT-related proteins. Further studies are required to confirm that *SPANXN2* suppresses tumor cell migration and colony formation by regulating EMT and AKT signaling pathways. Our results indicated that *SPANXN2* is a potential therapeutic target for TGCTs and will facilitate accurate and effective treatment of TGCTs.

### Funding

This research was funded by the National Natural Science Foundation of China (Grant No. 31472054), the National Key Research and Development Program of China (Nos. 2016YFC1000600), the Fundamental Research Funds for the Central Universities of Central South University (Nos. 2019zzts715), and the Fundamental Research Funds for Health Commission of Hunan Province(Grant Number: C2019073). The funders had no role in study design, data collection and analysis, decision to publish, or preparation of the manuscript.

### Grant Disclosures

The following grant information was disclosed by the authors:
National Natural Science Foundation of China: 31472054.
National Key Research and Development Program of China: 2016YFC1000600.
Fundamental Research Funds for the Central Universities of Central South University: 2019zzts715.
Fundamental Research Funds for Health Commission of Hunan Province: C2019073.

### Competing Interests

The authors declare there are no competing interests.

### Author Contributions

- Fang Zhu conceived and designed the experiments, performed the experiments, analyzed the data, prepared figures and/or tables, authored or reviewed drafts of the paper, and approved the final draft.
- Hao Bo conceived and designed the experiments, performed the experiments, authored or reviewed drafts of the paper, and approved the final draft.
- Guangmin Liu and Ruixue Li analyzed the data, prepared figures and/or tables, and approved the final draft.
- Zhizhong Liu performed the experiments, authored or reviewed drafts of the paper, and approved the final draft.

- Liqing Fan conceived and designed the experiments, authored or reviewed drafts of the paper, and approved the final draft.

## Human Ethics

The following information was supplied relating to ethical approvals (i.e., approving body and any reference numbers):

All patients provided written informed consent to tissue sample collection. This study was carried out with the approval and under the supervision of Ethic Committee of Basic Medical Science School, Central South University (Approval No: LLSB-2017-002).

## DNA Deposition

The following information was supplied regarding the deposition of DNA sequences:

The SPANX2 sequence is available at GenBank: NM_001009615.2.

## Data Availability

Raw data and pictures are available in the Supplemental Files.

## Supplemental Information

Supplemental information for this article can be found online at http://dx.doi.org/10.7717/peerj.9358#supplemental-information.

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
