# Peer review of "SPANXN2 functions a cell migration inhibitor in testicular germ cell tumor cells"

_PeerJ, doi:10.7717/peerj.9358_

## Round 0.1 · original submission · Major Revisions

All three reviewers have highlighted major concerns regarding this work. All of these must be addressed prior to any resubmission. These are extensive, and will require considerably more experimental work, but nonetheless I felt that you should have the opportunity to address these points, in full.

In particular note the comments regarding the use of transfected cell mixed populations: this must be addressed as the reviewers note that interpretation of your data in this context is difficult.

If you chose to address these points, given that the data will be essentially new, it is important for you to understand that this will then require a further round of rigorous peer-review.

I hope you find these comments of use.

Reviewer 1 ·

Basic reporting

1. Throughout the manuscript and especially in the introduction the use of English language should be significantly improved Examples lines 40-41, 52-54, 72-73…

Experimental design

For the referee, the information contained in the introduction, was not sufficient to understand the reasoning of the authors for choosing to investigate SPANX2 in TGCT tumors. Specifically, authors claim that it is important to understand the molecular signal of tumor metastasis (abstract) and then subsequently focus on the factor that is down regulated in the tumors vs normal tissue to discover that overexpression of this factor in tumor cell lines actually deteriorates tumor migration. It is therefore very difficult to imagine what role, if any, SPANXN2 plays in the metastasis of TGCT tumors. Maybe there are TGCT tumors that express different levels of Spanx2 that are correlated to survival/metastsis, but referee is not aware of this being the case.

Validity of the findings

The assays performed in this manuscript are difficult to interpret given the fact that the transient transfections are used. Cell cycle analyses after transient transfection with efficiency of 80% do not necessarily reflect the phenotype of Spanx2 overexpressing cells as this is a mixed population containing a significant amount of wt cells. Authors should have at best generated clonal cell lines that stably overexpress SPAnx2 and analyse per cell line two independent clones. Alternatively, the authors should have at least sorted the red fluorescent cells prior to cell cycle analyses. The same holds true for the colony assays performed. How do the authors know that colonies have grown from the cells expressing over expressing SPANX2? Because of this it very difficult to be sure that the data is interpreted correctly.
Furthermore for cell cycle analyses it states that experiment was repeated 2 times (line:146-147) in triplicate. In such a case graph 2E-F should not contain error bars, but rather two individual measurements. There are more examples of such inconsistencies, that referee will not detail at this point, but they shoudl all be corrected.

·

Basic reporting

No comment.

Experimental design

Please have a look at Major concerns: 3 to 6.

Validity of the findings

No comment.

Additional comments

Testicular germ cell tumors (TGCT) are the most frequently diagnosed solid tumors in young men and their incidence has been increasing over the past decades. Despite the relatively good prognosis and known etiopathogenesis of these tumors, neither targeted therapy nor molecular prognostic/predictive factors have yet been implemented in the management of TGCT, because there is not enough information about the molecular pathways or molecules involved in TGCT development that could be used for patient stratification and treatment. It is inspiriting and beneficial to analyze the molecular signals of metastases associated with this process. In this manuscript, Dr. Zhu and colleagues looked at the role of the SPANXN2 in diagnosis and prognosis of TGCT. The study’s conclusion showed that SPANXN2 inhibits the ability of colony formation and could be a tumor suppressor through EMT/AKT signaling pathways in TGCT. Although the current study is interesting, there are some major concerns that need to be addressed before consideration of publication.

Major concerns:
1. “Materials & Methods – Human Testicular Samples” Line 77-85. Please give the study approval number from medical ethics committee.
2. “Materials & Methods – Cell Culture” Line 103. Please provide references on the source of TCAM-2 cells.
3. “Results – Validation of SPANXN2 Expression in TGCTs” Line 185-191. “Subsequent quantitative RT-PCR analysis of TGCTs samples and normal samples also confirmed the differential expression of SPANXN2 (Figure 1B).” The mRNA levels alone are not sufficient to support the conclusion that SPANXN2 is downregulated in TGCTs, thus protein levels of SPANXNA2 should to be tested by Western blotting (WB) or Immunohistochemistry (IHC).
4. “Results – SPANXN2 Inhibited Colony Formation ability” Line 194-199. “Quantitative RT-PCR was used to detect the relative expression of SPANXN2 in these two transfected TGCTs cell lines (TCAM-2, NCCIT), using β-actin as a reference (Figure 2C, 2D).” Protein levels of SPANXNA2 should to be tested by Western blotting (WB) in SPANXNA2 overexpressed cells.
5. “Results – SPANXN2 Negatively Regulates TGCTs Cell Migration” Line 217-221. The conclusion that SPANXN2 overexpression inhibits TGCT cell migration is the most important finding of this study, thus more experiments on cell migration are needed to present, such as wound healing assay.
6. “Results – SPANXN2 Influences the EMT and AKT Signaling Pathways” Line 224-231. “These results revealed that the expression of the EMT proteins Vimentin, Snail and Claudin1 was decreased (Figure 4C), suggesting that SPANXN2 regulates the EMT signaling pathway.” The process of EMT is more of a transformation of cell morphology. Does TGCTs cells change in morphology occur in SPANXN2 overexpression? The author should give typical photos of cells.
7. “Results – SPANXN2 Influences the EMT and AKT Signaling Pathways” Line 224-231. EMT and AKT protein are not directly related to each other. More studies have shown that AKT signaling pathway is primarily involved in cell survival and proliferation [1-2]. What are functions of SPANXN2 in influencing the AKT signaling pathway?
[1] Song M, Bode AM, Dong Z, Lee MH. AKT as a Therapeutic Target for Cancer. Cancer Res. 2019;79(6):1019–1031. doi:10.1158/0008-5472.CAN-18-2738
[2] Fresno Vara JA, Casado E, de Castro J, Cejas P, Belda-Iniesta C, González-Barón M. PI3K/Akt signalling pathway and cancer. Cancer Treat Rev. 2004;30(2):193–204. doi:10.1016/j.ctrv.2003.07.007

Minor comments:
1. “Figure 1A and B”. Missing the units of y-axis label.
2. “Figure 2B”. Please use absolute numbers directly in clone formation analysis.
3. “Figure 2E and F”. Please perform the information of software and method for analyzing cell cycle. In addition, the scatter plot of flow cytometry should be perform for distribution of the cell cycle.

·

Basic reporting

The authors demonstrated that the expression of SPANXN2 was down-regulated in TGCTs compared with normal tissue. They also showed that SPANXN2 inhibited the migration of TGCTs cells through the down-regulation of EMT related genes Snail and Vimentin. However, there are several points to be clarified to confirm study conclusion.

Experimental design

Experimental design had no problem.

Validity of the findings

Because of the bad quality of Western Blot for Claudin 1 protein, authors cannot lead to the conclusion.

Additional comments

1. In Figure 4, the detection of Claudin 1 protein is unclear by Western blot. Also, it is unclear that AKT and p-AKT proteins are down-regulated by SPANXN2. The AKT and p-AKT protein levels should be quantified normalized by GAPDH.
2. The term, EMT/AKT pathway is not common. The tile should be corrected.

---

## Round 0.2 · Major Revisions

As you will see, reviewer-2 still has concerns which need to be addressed.

Each of these is important and must be carefully addressed and explained.

I also note that despite the overall recommendation, reviewer-1 remains unconvinced by your response to the issue of transient transfection approaches.

I share this view.

At a minimum, this issue must be carefully discussed in a revised version of the paper, explaining how this could negatively impact the fundings. This lack of stable cell clonal analysis does weaken the study in my view, but if this is carefully and clearly elaborated in the Discussion of the data I am prepared to let the field decide on the value of the work.

I enclose here an annotated PDF - the English still needs more work. I have gone through the abstract and the Intro to provide some direction as to how this should look. I hope you find this useful.

Reviewer 1 ·

Basic reporting

The use of English language has been somewhat improved even though still not perfect it is sufficient. The rationale of the study has been better illuminated.

Experimental design

My comments to experimental design still stand and have not been fully addressed. I do acknowledge replay by the authors in not being able to perform sorting of transfected cells. The hesitation in constructing the stable transfected cell line(s) is less understandable. It is true that during the process of generating such lines a certain adaptation to prolonged culture could take place, but that occurs in every (cancer)cell line during prolonged culture. This means that cell lines the authors are using for transient transfection differ at passage 10 and 20. Typically two clones of stably transfected should be generated and experimentally investigated, passages should be noted... However it is true that the others have been granted basing their conclusions on transient transfections.

Validity of the findings

no additional comment

Additional comments

no additional comments

·

Basic reporting

No comment.

Experimental design

No comment.

Validity of the findings

No comment.

Additional comments

Testicular germ cell tumors (TGCT) are the most frequently diagnosed solid tumors in young men and their incidence has been increasing over the past decades. Despite the relatively good prognosis and known etiopathogenesis of these tumors, neither targeted therapy nor molecular prognostic/predictive factors have yet been implemented in the management of TGCT, because there is not enough information about the molecular pathways or molecules involved in TGCT development that could be used for patient stratification and treatment. It is inspiriting and beneficial to analyze the molecular signals of metastases associated with this process. In this manuscript, Dr. Zhu and colleagues looked at the role of the SPANXN2 in diagnosis and prognosis of TGCT. The study’s conclusion showed that SPANXN2 inhibits the ability of colony formation and could be a tumor suppressor through EMT/AKT signaling pathways in TGCT. There are still several major concerns that need to be addressed, although the author has answered my question point to point.

Major concerns:
1. “The reply of major concerns 3”. I have found an antibody (SPANX-N2 MaxPab mouse polyclonal antibody, Catalog #: H00494119-B01, Abnova, http://www.abnova.com/products/products_detail.asp?catalog_id=H00494119-B01 ) that can detect the level of SPANXN2 with Western Blotting, so please check the expression of SPANXN2 in tissues and cells.
2. “The reply of major concerns 4”. In this manuscript, Line 117-118. “The sequence of SPANXN2 was cloned into the CMV-MCS-DsRed2-SV40-Neomycin-GV147 vector.” In the overexpression vector, DsRed2 is the C-terminal directly connected to SPANXN2. Although red fluorescence was observed after cell transfection, the translation of SPANXN2 was not completely guaranteed to be normal. Therefore, in rigorous experimental studies, the mRNA and protein levels of SPANXN2 should be further tested to ensure the success of the experiment.
3. “The reply of major concerns 5”. “We had performed the wound healing assay, but the difference of the result was not statistically significant between NC and SPANXN2. This may be due to the influence of cell proliferation is hard to exclude in the wound healing assay or other factors.” This sentence makes me feel very confused! If overexpression of SPANXN2 can inhibit both proliferation and migration, the results of the wound healing experiment should be significant rather than undifferentiated. If overexpression of SPANXN2 does not affect proliferation but does affect cell migration, then the wound healing assay results should be consistent with the transwell migration results. Thus, please present the results and give a further explanation. In addition, except for the wound healing assay and transwell migration assay, there are other experiments that can reflect the cell movement or migration.

---

## Round 0.3 · accepted · Accept

Thank you for addressing the points raised in the last review cycle. As I indicated, although I would prefer this work to use stable transfected cell lines, your description of the limitation of work means that the field can decide for itself how it views these observations. Hence, given the attention to the other points which you have provided, I am happy to recommend acceptance. Congratulations.